# Ecological indices of phytophagous Hemiptera and their natural enemies on *Acacia auriculiformis* (Fabales: Fabaceae) plants with or without dehydrated sewage sludge application in a degraded area

**Luan Rocha Dourado**[1], **Germano Leão Demolin-Leite**[2]*, **Marcus Alvarenga Soares**[1], **Gustavo Leal Teixeira**[2], **Farley William Souza Silva**[3], **Regynaldo Arruda Sampaio**[2], **Jose Cola Zanuncio**[4], **Jesusa Crisostomo Legaspi**[5]

1 Departamento de Agronomia, Universidade Federal dos Vales do Jequitinhonha e Mucuri, Diamantina, Minas Gerais, Brazil, 2 Instituto de Ciências Agrárias, Universidade Federal de Minas Gerais, Montes Claros, Minas Gerais, Brazil, 3 Departamento de Agronomia, Universidade Federal do Acre, Rio Branco, Acre, Brazil, 4 Departamento de Entomologia/BIOAGRO, Universidade Federal deVicosa, Vicosa, Minas Gerais, Brazil, 5 United States Department of Agriculture-Agricultural Research Service-Center for Medical, Agricultural and Veterinary Entomology, Tallahassee, Florida, United States of America

* germano.demolin@gmail.com

## Abstract

Soil fertilization with dehydrated sewage sludge (DSS) accelerates the recovery process of degraded areas by improving nutrient concentration, and favors the development of trophic webs with pioneer plants such as *Acacia auriculiformis* A. Cunn. ex Beth (Fabales: Fabaceae), phytophagous Hemiptera, predators, and protocooperanting ants. This study aimed to evaluate the development and production of *A. auriculiformis* litter with or without dehydrated sewage sludge application and the ecological indices of sucking insects (Hemiptera), their predators and protocooperating ants, as bioindicators, in a degraded area for 24 months. Complete randomization was applied for two treatments (with or without application of dehydrated sewage sludge) in 24 replications (one repetition = one plant). We evaluated the number of leaves/branch and branches/plant, percentage of soil cover (litter), ecological indices of phytophagous Hemiptera, their predators, and protocooperating ants. The plants of *A. auriculiformis*, that were applied with dehydrated sewage sludge, had superior development when compared to plants where DSS were not applied. The highest abundance and richness of phytophagous Hemiptera species and Sternorrhyncha predators occurred on *A. auriculiformis* plants that were applied with dehydrated sewage sludge. The increase in richness of species of protocooperanting ants that established mutualistic relationships positively influenced the phytophagous Hemiptera. The use of *A. auriculiformis*, with application of dehydrated sewage sludge, can increase recovery of degraded areas due to its higher soil cover (e.g., litter) and results in higher ecological indices of phytophagous Hemiptera and their predators.

**Data Availability Statement:** All relevant data are within the manuscript and its Supporting Information files.

**Funding:** The study was financially supported by the following Brazilian agencies "Conselho Nacional de Desenvolvimento Científico e Tecnológico (CNPq)", "Fundação de Amparo à Pesquisa do Estado de Minas Gerais (FAPEMIG)", and "Programa 23 Cooperativo sobre Proteção Florestal (PROTEF)" of the "Instituto de Pesquisas e Estudos Florestais (IPEF).

**Competing interests:** The authors have declared that no competing interests exist.

## Introduction

Tropical soils (eg., Brazil), in general, are highly weathered, have poor chemical quality and fragile macrostructure [1]. Moreover, the tropical climate, with high temperatures and humidity, accelerates the degradation of soil organic matter [2]. These factors, combined with poor soil management, forest clearing and burning, intensive mechanization, and grazing, promote changes in ecosystems at a faster rate than the natural regeneration capacity [2]. A destabilized ecosystem, in turn, negatively affects species richness, abundance, and distribution [3]. Therefore, depending on the intensity of soil degradation, the use of restoration techniques is suggested for fauna and flora rehabilitation [4].

In this context, species of the genus *Acacia* (Fabales: Fabaceae) may be useful due to their rapid growth and the capacity for biological nitrogen fixation (BNF) in association with symbiotic bacteria [5]. The natural introduction of nitrogen can intensify the cycle of other nutrients and stabilize soil organic matter in degraded environments [6]. Among the species of this genus, *Acacia auriculiformis* A. Cunn. ex Beth stands out for its resilience, lower susceptibility to disease, and adaptability [7]. *A. auriculiformis* also provides other ecosystem services such as moisture retention, potassium deposition, soil organic carbon (litter) and heavy metal phytoextraction with mycorrhizal associations [8, 9].

Among the varieties of waste produced by anthropological activity, sewage sludge production stands out as a by-product of urban wastewater treatment facilities [10]. In Brazil, dehydrated sewage sludge is used in agriculture (e.g., *Saccharum* sp. L. (Poaceae) and *Phaseolus vulgaris* L. (Faboideae)) and in reforestation (e.g., *Acacia mangium* Willd. (Fabaceae) and *Pittosporum tenuifolium* Sol. Ex Gaertn. (Pittosporaceae)) as a fertilizer and soil conditioning agent [11–14]. As it holds significant amounts of organic matter and nutrients (e.g., nitrogen and phosphorus), sewage sludge improves plant growth and development and the physico-chemical, and biological properties of the soil [15]. However, due to the high concentration of nutrients, heavy metals and persistent organic pollutants, the inappropriate disposal of sewage sludge can cause environmental impacts [14, 16]. Also, sewage sludge, through its nutrients, can impact insect population where N levels are above or below ideal, affecting the physiology, diversity, and distribution of phytophagous insects [14]. The ecological indices of these species can be employed to monitor the recovery of degraded areas due to its great diversity, amount of occupied habitats, importance in biological processes and rapid response to environmental changes [17]. Insects of the orders Hemiptera (e.g., Cicadidae) and Hymenoptera (e.g., Formicidae), for instance, are used as bioindicators of degraded areas recovery [18, 19]. In this context, plants that grow vigorously are more susceptible to attacks of herbivorous insects—Plant Vigor Hypothesis—generating greater diversity and abundance of insects and therefore, natural enemies [20]. Under these conditions, the same ecological processes of the theory of biogeographic islands (BGI) apply to plants, with a higher probability of extinction of rarer species in smaller BGI [21].

Thus, the objective of this study was to evaluate the growth and development and ground cover by *A. auriculiformis*, with or without application of dehydrated sewage sludge, and ecological indices of phytophagous Hemiptera, Sternorrhyncha predators and protocooperating ants, as bioindicators, in a degraded area by testing two hypotheses: i) plants with application of dehydrated sewage sludge will have larger crowns and form more litter, thus assisting in the recovery of degraded soils and ii) plants with application of dehydrated sewage sludge will be larger (> BGI) and with better nutritional quality (> free amino acids), greater abundance, species richness and diversity of phytophagous Hemiptera and, consequently, Sternorrhyncha predators and protocooperating ants.

## Materials and methods

### Experimental site

The study was carried out in a degraded area at the "Instituto de Ciências Agrárias (ICA)" of the "Universidade Federal de Minas Gerais (UFMG)," Montes Claros city, Minas Gerais State, Brazil (latitude 16°51' S × longitude 44°55' W, altitude 943 m) from March 2017 to February 2019 (24 months; arthropod collection period). The area was defined as degraded due to soil losses and changes in soil chemistry or hydrology [22]. The climate of this area is found to be Aw: tropical savannah, with dry winters and rainy summers, according to the Köppen classification [23], with annual precipitation between 1000–1300 mm and a yearly average temperature of $\geq$ 28°C. The type of soil is litolic neosoil, loamy texture, total sand = 17 dag.Kg$^{-1}$, silt = 46.0 dag.Kg$^{-1}$, clay = 37.0 dag.Kg$^{-1}$, pH–$H_2O$ = 4.3, organic matter = 0.73 dag.Kg$^{-1}$, organic carbon = 0,42 dag.Kg$^{-1}$, P = 0.35 mg.dm$^{-3}$, K = 41.0 mg.dm$^{-3}$, Ca = 1.6 cmol$_c$.dm$^{-3}$, Mg = 0.9 cmol$_c$.dm$^{-3}$, Al = 3.3 cmol$_c$.dm$^{-3}$, aluminum saturation in the capacity of cationic exchange = 55.1%, sum of bases = 2.69 cmol$_c$ dm$^{-3}$, H + Al = 13.4 cmol$_c$.dm$^{-3}$, percentage of soil base saturation of the capacity of cationic exchange a pH 7.0 = 16.7, effective cation exchange capacity (CEC) = 5.9 cmol$_c$.dm$^{-3}$, and potential (pH 7.0) CEC = 16.1 cmol$_c$.dm$^{-3}$ [14].

### Experimental design

In March 2016, *A. auriculiformis* seeds were obtained from 5-year-old trees grown at ICA/ UFMG. *A. auriculiformis* seedlings were grown in a nursery in plastic bags (8 x 12 cm) with a substrate mixture of 30% organic compost, 30% clay soil, 30% sand, and 10% reactive natural phosphate (160 g)/pit. The organic compost consists of three parts by volume: two parts of chopped prunings ($\leq$ 5 cm) and one part of tanned manure. The soil pH in the pits was rectified with dolomitic limestone (relative total neutralization power of 90%) (187 g/pit), increasing the base saturation to 50% [24]. Natural phosphate (80g/pit), fritted trace elements (FTE) (10g/pit), and marble dust (1kg/pit) were added according to the soil quality. *A. auriculiformis* seedlings, with six month old, were transplanted 30 cm high in pits (40 × 40 × 40 cm) two meters apart, in six parallel lines on flat ground (same characteristics). In September 2016, 24 plants were treated with a single dose of 20 L dehydrated sewage sludge/pit and 24 plants were left untreated. The seedlings were irrigated twice a week until the beginning of the rainy season when no more water was provided. The plants were pruned with a sterilized razor (each plant) when their branches reached 5 cm in length, cutting the additional stems and branches up to 1/3 of crown height, leaving out only the best stem. All pruned parts of the plants were left between planting lines. The design was completely randomized with two treatments (with dehydrated sewage sludge and without sewage sludge) with 24 replications (one repetition = one plant).

Dehydrated sewage sludge (5% moisture) was collected at the "Estação de Tratamento de Esgoto (ETE)" in the city of Juramento, Minas Gerais, Brazil, about 40 km from the *A. auriculiformis* planting site. The main biochemical characteristics of the dehydrated sewage sludge of this company were: pH–$H_2O$ = 4.40, N = 10.4 mg.Kg$^{-1}$, P = 2.9 mg.Kg$^{-1}$, K = 5.8 mg.Kg$^{-1}$, Cd = 0.1 µg.g$^{-1}$, Pb = 56.9 µg.g$^{-1}$, Cr = 46.7 µg.g$^{-1}$, and fecal coliforms = 4.35 most likely number g$^{-1}$ [25, 26].

### Plant mass production and soil coverage

The numbers of leaves/branch and branches/plant of 48 *A. auriculiformis* plants, and percentage of soil cover by litter, grass and herbaceous plants below their crowns (plot 1.0 m$_2$) were evaluated visually every month.

## Insects

The insects were counted visually, fortnightly, between 7 and 11 am, on the adaxial and abaxial sides of leaves in the apical, middle and basal parts of the canopy and in the north, south, east and west directions, totaling 12 leaves/plant/evaluation, in each of the 48 six-month-old *A. auriculiformis* trees for 24 months. Insects were not removed from plants during evaluations. The total sampling effort was of 27648 leaves covering the entire plant (vertical and horizontal axes) for observation of as many insect species as possible, especially the rarest ones. At least three specimens per species of insects were captured by a vacuum cleaner, stored in 70% ethanol glass vials or assembled, broken down into morphospecies, and sent for identification (S1 File).

## Ecological indices

Ecological indices (species abundance, diversity, and species richness) were calculated for the species identified in the treatments (with or without dehydrated sewage sludge)/tree with Bio-Diversity Professional, Version 2 (© 1997 The Natural History Museum: http://www.sams.ac.uk/dml/projects/benthic/bdpro/index.htm) [27]. Diversity was calculated with Hill's formula [28] and species richness with the Simpson Index [29].

## Statistics

The leaves/branch, branches/plant data and percentage of soil cover per litter, herbaceous and grassy plants, abundance, diversity and species richness of Phytophagous Hemiptera, Sternorrhyncha predators and protocooperating ants were subjected to non-parametric statistical test, Wilcoxon signed-rank test (P $<0.05$) [30] by the System for Statistical and Genetic Analysis—SAEG, version 9.1 [31]. Data were analyzed using simple regression or principal component regression (PCR) when linear (P $<0.05$) to test the interactions between these groups of insects and *A. auriculiformis* total number of leaves and branches. The regression model known as PCR applies principal component analysis, based on a covariance matrix, to perform regression. Thus, it is possible to reduce the dimension of regression by the exclusion of the aspects that contribute to collinearity problems, or, linear relationships between the independent variables. All results were significant at (P $<0.05$) for variable selection based on the stepwise method. No specific permits are required to plant *Acacia auriculiformis* in Brazil. The laboratory and field studies did not involve endangered or protected species.

# Results

## Effect of dehydrated sewage sludge treatment on *A. auriculiformis* plants

The plants of *A. auriculiformis* treated with dehydrated sewage sludge had higher numbers of leaves/branch, branches/plant and percentage of soil cover (eg., litter) (P < 0.05) (Table 1).

## Insect ecological indexes

The highest abundance, diversity, and species richness (P < 0.05) of phytophagous Hemiptera and Sternorrhyncha predators occurred in dehydrated sewage sludge treated *A. auriculiformis* plants. However, the ecological indices of protocooperating ants did not differ statistically (P > 0.05) between the treatments. The increase of leaves/branch and branches/plant affected positively the species richness of phytophagous Hemiptera, abundance, species richness and species diversity of Sternorrhyncha predators, and the abundance of protocooperating ants. Enhancement in abundance and richness of protocooperating ants species positively influenced the same parameters of phytophagous Hemiptera, and vice versa (Tables 1 and 2).

**Table 1. Numbers of leaves/branch, branches/plant, percentage of soil cover, abundance (Ab.), diversity index (D.) and species richness (S.R.) of phytophagous Hemiptera (Hem.), protocooperating ants (Ants), and Sternor-rhyncha predators (Ster.Pred.) on *Acacia auriculiformis* plants (mean±SE) with and without application of dehydrated sewage sludge.**

|  | Dehydrated sewage sludge | | Test of Wilcoxon | |
|---|---|---|---|---|
|  | **With** | **Without** | ***VT**\* | ***P*** |
| Leaves/branch | 35.00±1.03 | 28.23±0.89 | 4.2 | 0.00 |
| Branches/plant | 50.10±1.29 | 24.74±0.60 | 5.9 | 0.00 |
| Percentage of soil cover | 29.34±1.25 | 6.87±0.42 | 3.5 | 0.00 |
| Ab.Hem. | 8.17±1.96 | 3.29±1.46 | 3.3 | 0.00 |
| D.Hem. | 4.99±0.79 | 3.12±0.43 | 1.6 | 0.04 |
| S.R.Hem. | 2.38±0.27 | 1.34±0.20 | 2.8 | 0.00 |
| Ab.Ants. | 18.29±3.30 | 11.33±1.36 | 1.2 | 0.12 |
| D.Ants | 6.84±0.64 | 6.53±0.61 | 0.3 | 0.37 |
| S.R.Ants | 3.30±0.32 | 3.21±0.24 | 0.6 | 0.27 |
| Ab.Ster.Pred. | 1.54±0.29 | 0.83±0.18 | 1.8 | 0.03 |
| D.Ster.Pred. | 3.38±0.59 | 1.44±1.44 | 2.9 | 0.00 |
| S.R.Ster.Pred. | 1.08±0.17 | 0.59±0.10 | 2.1 | 0.02 |

$VT^*$ = value of test. n = 24 per treatment.

## Discussion

*A. auriculiformis* plants treated with dehydrated sewage sludge, had higher crowns, with an increase in leaves ($>$ 6.7) and branches ($>$ 12.2) per plant. This confirms the first hypothesis that dehydrated sewage sludge treated plants will be larger and with higher litter deposition, which enhances the recovery process of degraded areas [32, 33]. Application of dehydrated sewage sludge in degraded soil provides better conditions for the growth and development of *A. auriculiformis* [14]. Similar observations have been reported in *A. mangium* Willd. (Fabaceae), *Cordyline australis* (G. Forst.) Endl (Asparagaceae), *Eucalyptus grandis* Hill (Myrtaceae), *Lafoensia pacari* Saint-Hilaire (Lythraceae) and *Pittosporum tenuifolium* Sol. Eg. Gaertn. (Pittosporaceae) [13, 14, 34–36]. Therefore, it can be concluded that application of dehydrated sewage sludge as fertilizer in the degraded areas can accelerate the fertility recovery process which is normally a slow process [37]. Besides, *A. auriculiformis* is promising in the recovery

**Table 2. Relationships between abundance (Ab.) of protocooperating ants (Ants), Sternorrhyncha predators (Ster.Pred.) and phytophagous Hemiptera (Hem.); diversity (D.) of Ster.Pred.; species richness (S.R.) of Hem. and Ster.Pred. with leaves/branch (Nleaves) and/or branches/plant (Nbranches) and S.R. numbers of Hem. with S.R. of ants on *Acacia auriculiformis* plants.**

| Equations of principal component regression | ANOVA | | |
|---|---|---|---|
|  | **R²** | ***F*** | ***P*** |
| Ab.Ants = -3.85+0.52xNleaves+0.40xAb.Hem. | 0.16 | 4.12 | 0.02 |
| S.R.Hem. = -0.70+0.38xS.R.Ants+0.04xNbranches | 0.32 | 10.32 | 0.00 |
| S.R.Ster.Pred. = 0.18+0.02xNbranches | 0.10 | 5.14 | 0.02 |
| D.Ster.Pred. = -0.81+0.09xNbranches | 0.60 | 23.59 | 0.00 |
| Equations of simple regression analysis |  |  |  |
| Ab.Ster.Pred. = 0.01+0.001xNleaves | 0.13 | 6.63 | 0.01 |
| D.Ster.Pred. = 28.22–1.85xNleaves+0.03xNleaves² | 0.58 | 18.22 | 0.00 |
| S.R.Ster.Pred. = 7.42–0.44xNleaves+0.08xNleaves² | 0.13 | 3.57 | 0.04 |

n = 48, degrees of freedom: treatment = 1, repetitions = 22, and residue = 23.

of degraded areas due to its pioneering nature, which guarantees assistance for the development of other species [7–9]. A treated sewage sludge from the "Estação de Tratamento de Esgoto (ETE)", Juramento municipality, Minas Gerais State, Brazil, had no helminth eggs and protozoan cysts, and did not increase the heavy metal contents in grains of *Zea mays* L. (Poales: Poaceae) and *Vigna unguiculata* (L.) Walp. (Fabales: Fabaceae) [26].

*A. auriculiformis* plants planted in degraded plots with dehydrated sewage sludge, had higher ecological indices of phytophagous Hemiptera, Sternorrhyncha predators, leaves and protocooperating ants. Enhancement occurs due to larger canopy sizes confirms the second hypothesis that dehydrated sewage sludge treated plants will be larger and with greater abundance, species richness and diversity of phytophagous Hemiptera and, consequently, their predators. Each plant alone behaves as a small-scale BGI, where those that grow rapidly and reach larger than average size are preferred by herbivores and are also subject to more complex plant-arthropod interactions (i.e., plant vigor hypothesis) [20, 21]. Additional studies can prove the positive correlation between phytophagous Hemiptera and the boost of plant crown (> biomass> resources), for example, the abundance of phytophagous insects in *A. mangium* Wild. (Fabales: Fabaceae); galling insects in *Macairea radula* (Bonpl.) DC. (Myrtales: Melastomataceae); and *Carpatolechia proximella* Hbn. (Lepidoptera: Gelechiidae) in *Picea abies* (L.) Karst. (Pinales: Pinaceae) [14, 38–40]. On the other hand, the increase in nutrient availability, especially nitrogen, provided by application of dehydrated sewage sludge, reflects upon the quality of sap (> amount of protein and free amino acids) [41, 42]. This increase benefits sucking insects that get enough nutrients to survive through the host plant sap (e.g., Sternorrhyncha), improving their performance and population density [43]. Also, larger plants offer avoidance from enemies due to greater size and architectural complexity, reflecting on the distribution of herbivorous insects [44, 45]. The increase in habitat complexity by larger plants also provides indirect benefits to natural enemies, as it supports greater abundance of phytophages and increases the chances of rare species maintenance [21, 46]. Thus, higher BGI affects a more significant number of predators in response to their prey abundance and lifestyle [47]. Also, predatory insects generally have smaller population sizes than their prey; therefore, they must face a higher probability of local extinction, particularly in smaller plants (<BGI) [48]. Some ant species establish symbiosis with numerous Sternorrhyncha species [49]. In places where ants feed on honeydew—sugary substances secreted by carbohydrate-rich aphids offer protection against natural enemies, also frightening other competing phytophages [50]. Thus, the presence of protocooperating ants reduces predators, competitors and encourages the presence of other species of Sternorrhyncha [51].

## Conclusions

*A. auriculiformis* plants grown in plots treated with dehydrated sewage sludge have higher crowns, resulting in increased litter deposition and helping the recovery of degraded soils. These plants show larger BGI and, consequently, greater abundance, species richness and diversity of phytophagous Hemiptera and Sternorrhyncha predators.

## Supporting information

**S1 File. Species of phytophagous Hemiptera, Sternorrhyncha predators, and protocooperating ants on Acacia auriculiformis plants.**
(DOCX)

**S1 Data.**
(XLS)

## Acknowledgments

We would like to thank to Dr. Carlos Matrangolo (UNIMONTES) (Formicidae), Dr. Ivan Cardoso Nascimento (EMBRAPA-ILHÉUS) (Formicidae), Dr. Luci Boa Nova Coelho (UFRJ) (Cicadellidae) and Dr. Paulo Sérgio Fiuza Ferreira (UFV) (Hemiptera) for specimen identifications. Voucher numbers are 1595/02 and 1597/02 (CDZOO-UFPR).

## Author Contributions

**Conceptualization:** Germano Leão Demolin-Leite, Regynaldo Arruda Sampaio.

**Data curation:** Luan Rocha Dourado, Marcus Alvarenga Soares, Farley William Souza Silva.

**Formal analysis:** Germano Leão Demolin-Leite, Marcus Alvarenga Soares.

**Funding acquisition:** Regynaldo Arruda Sampaio.

**Investigation:** Luan Rocha Dourado, Germano Leão Demolin-Leite, Farley William Souza Silva.

**Methodology:** Germano Leão Demolin-Leite, Regynaldo Arruda Sampaio.

**Supervision:** Regynaldo Arruda Sampaio.

**Writing – original draft:** Luan Rocha Dourado, Germano Leão Demolin-Leite, Marcus Alvarenga Soares, Gustavo Leal Teixeira, Farley William Souza Silva, Regynaldo Arruda Sampaio.

**Writing – review & editing:** Jose Cola Zanuncio, Jesusa Crisostomo Legaspi.

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
