## [Decision Letter · Decision Letter 0]

25 Jun 2020

PONE-D-20-08541

Can Acacia auriculiformis (Fabales: Fabaceae) fertilized with dehydrated sewage sludge contribute to the regeneration of degraded areas?

PLOS ONE

Dear Dr. Demolin Leite

Thank you for submitting your manuscript to PLOS ONE. After careful consideration, we feel that it has merit but does not fully meet PLOS ONE’s publication criteria as it currently stands. Therefore, we invite you to submit a revised version of the manuscript that addresses the points raised during the review process.

We look forward to receiving your revised manuscript.

Kind regards,

Tunira Bhadauria, Ph.D.

Academic Editor

PLOS ONE

Journal Requirements:

2. Thank you for stating the following above the Acknowledgments Section of your manuscript:

'Funding. The study was financially supported by the following Brazilian agencies “Conselho Nacional de Desenvolvimento Científico e Tecnológico (CNPq)”, “Coordenação de  Aperfeiçoamento de Pessoal de Nível Superior (CAPES-Finance Code 001)”, “Fundação de Amparo à Pesquisa do Estado de Minas Gerais (FAPEMIG)”, and “Programa Cooperativo  sobre Proteção Florestal (PROTEF)” of the “Instituto de Pesquisas e Estudos Florestais (IPEF)”.'

'NO - Include this sentence at the end of your statement: The funders had no role in study design, data collection and analysis, decision to publish, or preparation of the manuscript.'

Additional Editor Comments:

The present paper has interesting research aspects specially for utilizing dehydrated sewage sludge for sustainable use in restoration and rehabilitation of degraded lands. However after going through the reviewers comments I think both have raised very valid comments and suggestions which I endorse fully, each of these comments and suggestions need to be incorporated into the manuscript before it is accepted for the publication

I suggest that the authors answer each of the comments pointwise as bullets, for each of the two reviewers separately. Therefore I recommend a major revesion of the manuscript before the same is accepted for the publication.

Reviewers' comments:

Reviewer's Responses to Questions

**Comments to the Author**

1. Is the manuscript technically sound, and do the data support the conclusions?

Reviewer #1: Partly

Reviewer #2: Yes

2. Has the statistical analysis been performed appropriately and rigorously? 

Reviewer #1: Yes

Reviewer #2: Yes

3. Have the authors made all data underlying the findings in their manuscript fully available?

Reviewer #1: No

Reviewer #2: No

4. Is the manuscript presented in an intelligible fashion and written in standard English?

Reviewer #1: Yes

Reviewer #2: Yes

5. Review Comments to the Author

Reviewer #1: The comments related to manuscript"Can Acacia auriculiformis (Fabales: Fabaceae) fertilized with dehydrated sewage sludge contribute to the regeneration of degraded areas? " are as follows:

1. Title itself is confusing as it appears the plant A.auriculiformis is treated with dehydrated sewage sludge which is not the case its the soil which has been treated with dehydrated sewage sludge.

2.Soil aspects like soil nutrient quality before and after application of dehydrated sewage sludge in the degraded plots is lacking.

3. Too many references are included in the manuscript, that must be reduced.

4. Authors have not followed the format of the Journal.

5.Fertilized and Fertilization words are confusing . consider changing it with application of or treatment with dehydrated sewage sludge

6. soil quality assessment is very necessary while studying with regeneration/ restoration of degraded areas.

Reviewer #2: Dear Authors,

I would congratulate you for this interesting research work and the paper written especially for using dehydrated sewage sludge that is of important concern in the times when population and sewage load is increasing globally and how it can be sustainably used in restoration or rehabilitation of degraded lands. Especially in light of UN decade of restoration being declared from 2021-2030 these inputs and ideas will be of concern to enhance sustainable land restoration across the world.

I have few general and some specific comments that you would like to address and incorporate to improve the ms.

General:

1. The flow of ms under different heading was confusing as Material and Methods section was after results and discussion section that ideally should be after introduction and before results and discussion. Please see author guidelines

2. Please share a few images of the experimental design as well as different insect species observed during the study for better clarity of the experiment.

Specific Comments:

Though, the ms is an interesting approach that needs to be published here are my few comments that I think are missing in the present form.

1. Line 66-68 mentions about the inappropriate disposal of sewage sludge and later briefly 191-192 mentions about the sewage sludge chemical and biological characteristics. It is important that soil physico-chemical and biological characteristics along with sewage sludge physico-chemical and biological characteristics are mentioned in a Table in the ms for better clarity and understanding of the readers.

2. Line 194-195 in the material and methods section talks about grass and herbaceous plant being evaluated visually per month but there is no result provided about this how diverse herbs and grass species have grown throughout this experiment and it has not been discussed in the discussion section.

3. Because this was a long term experimental work it would have been interesting to know the soil physico chemical and biological properties of the soil improved with and without sewage sludge treatment. This is specially important for long term consideration to use sewage sludge for restoration of degraded land.

4. Because sewage sludge may be the source of heavy metals and persistent organic pollutants counting diverse insects should not be only approach it should be well considered that during this entire process their was no heavy metal uptake by plants. For this it is better to address suggestion as mentioned in point 1 and if there was certain amount of heavy metal present it is important to also monitor the heavy metal uptake in plant. As Acacia is fast going plant it can bioaccumulate heavy metal in biomass.

5. It may be the case that for this particular study sewage sludge is devoid of any heavy metal concentration but those who would like to replicate this approach it is important that this caution is well discussed in the discussion section.

6. PLOS authors have the option to publish the peer review history of their article (what does this mean?). If published, this will include your full peer review and any attached files.

Reviewer #1: **Yes: **Dr. Anil Kumar

Reviewer #2: No

---

## [Author Response · Author response to Decision Letter 0]

13 Jul 2020

Dear Academic Editor, Plos One

Dr. Tunira Bhadauria

We are sending attached a corrected version of the manuscript entitled “Ecological indices of phytophagous Hemiptera and their natural enemies on Acacia auriculiformis (Fabales: Fabaceae) plants with or without dehydrated sewage sludge application in a degraded area” code PONE-D-20-08541, reviewed by the authors (in red colour), to continue its evaluation for possible publication as a Research Article in Plos One.

We received valuable help from the two specialists: Dr. Jose Cola Zanuncio (Brazil) and Dr. Jesusa Crisostomo Legaspi (USA), and both to conduct the requested corrections in the manuscript suggested by the Reviewer #1 and Reviewer #2. We have added Dr. J.C. Zanuncio and Dr. J.C. Legaspi as the penultimate and ultimate authors, respectively, after approval from all authors.

Please, put in my name, the following in your system: Funding. The study was financially supported by the following Brazilian agencies “Conselho Nacional de Desenvolvimento Científico e Tecnológico (CNPq)”, “Fundação de Amparo à Pesquisa do Estado de Minas Gerais (FAPEMIG)”, and “Programa Cooperativo sobre Proteção Florestal (PROTEF)” of the “Instituto de Pesquisas e Estudos Florestais (IPEF)”.

We are really pleased with the positive comments our manuscript has attracted from the Reviewer #1 and Reviewer #2. It is worth mention that all comments were addressed and helped us to improve the manuscript quality.

Below we describe how we have addressed point-by-point the issues rose by the Reviewer #1 and Reviewer #2 (original Reviewer comments in regular typeface, responses in red). We believe that our manuscript has achieved the Plos One quality standards, and we look forward to receive your decision.

Your sincerely, 

Prof. Dr. Germano Leão Demolin Leite

Reviewers' comments:

Reviewer's Responses to Questions

Comments to the Author

1. Is the manuscript technically sound, and do the data support the conclusions?

Reviewer #1: Partly

Reviewer #2: Yes

Author: We realized the corrections.

2. Has the statistical analysis been performed appropriately and rigorously?

Reviewer #1: Yes

Reviewer #2: Yes

3. Have the authors made all data underlying the findings in their manuscript fully available?

Reviewer #1: No

Reviewer #2: No

Author: We will put our data on Plos One site.

4. Is the manuscript presented in an intelligible fashion and written in standard English?

Reviewer #1: Yes

Reviewer #2: Yes

5. Review Comments to the Author

Reviewer #1: The comments related to manuscript"Can Acacia auriculiformis (Fabales: Fabaceae) fertilized with dehydrated sewage sludge contribute to the regeneration of degraded areas? " are as follows:

1. Title itself is confusing as it appears the plant A. auriculiformis is treated with dehydrated sewage sludge which is not the case its the soil which has been treated with dehydrated sewage sludge.

Author: We realized the corrections.

2.Soil aspects like soil nutrient quality before and after application of dehydrated sewage sludge in the degraded plots is lacking.

Author: We put, in this new version, the information (in text form) of the soil physico-chemical characteristics. But we do not have after application of the fertilizer.

3. Too many references are included in the manuscript, that must be reduced.

Author: We cut several references (87→51) 

4. Authors have not followed the format of the Journal.

Author: We followed the formato of the journal in this version.

5.Fertilized and Fertilization words are confusing . consider changing it with application of or treatment with dehydrated sewage sludge

Author: We used now the “with application of dehydrated sewage sludge”

6. soil quality assessment is very necessary while studying with regeneration/ restoration of degraded areas.

Author: We put, in this new version, the information (in text form) of the soil physico-chemical characteristics. But we do not have after application of the fertilizer.

Reviewer #2: Dear Authors,

I would congratulate you for this interesting research work and the paper written especially for using dehydrated sewage sludge that is of important concern in the times when population and sewage load is increasing globally and how it can be sustainably used in restoration or rehabilitation of degraded lands. Especially in light of UN decade of restoration being declared from 2021-2030 these inputs and ideas will be of concern to enhance sustainable land restoration across the world.

Author: Thank you for your words.

I have few general and some specific comments that you would like to address and incorporate to improve the ms.

General:

1. The flow of ms under different heading was confusing as Material and Methods section was after results and discussion section that ideally should be after introduction and before results and discussion. Please see author guidelines

Author: We followed the formato of the journal in this version.

2. Please share a few images of the experimental design as well as different insect species observed during the study for better clarity of the experiment.

Author: We do not have photos, sorry.

Specific Comments:

Though, the ms is an interesting approach that needs to be published here are my few comments that I think are missing in the present form.

1. Line 66-68 mentions about the inappropriate disposal of sewage sludge and later briefly 191-192 mentions about the sewage sludge chemical and biological characteristics. It is important that soil physico-chemical and biological characteristics along with sewage sludge physico-chemical and biological characteristics are mentioned in a Table in the ms for better clarity and understanding of the readers.

Author: We put, in this new version, the informations (in text form) of the soil physico-chemical (we do not have biological characteristic) and of the sewage sludge physico-chemical and biological characteristics.

2. Line 194-195 in the material and methods section talks about grass and herbaceous plant being evaluated visually per month but there is no result provided about this how diverse herbs and grass species have grown throughout this experiment and it has not been discussed in the discussion section.

Author: the percentage of soil cover by litter, grass and herbaceous plants were evaluated visually per month and plot (1.0 m2), but we did not separate litter of plants, total coverage was done, in percentage of ground cover. We discussed about ground cover (e.g. litter).

3. Because this was a long term experimental work it would have been interesting to know the soil physico chemical and biological properties of the soil improved with and without sewage sludge treatment. This is specially important for long term consideration to use sewage sludge for restoration of degraded land.

Author: We put, in this new version, the informations (in text form) of the soil physico-chemical (we do not have biological characteristic) and of the sewage sludge physico-chemical and biological characteristics. But we do not have data after application of this fertilizer. 

4. Because sewage sludge may be the source of heavy metals and persistent organic pollutants counting diverse insects should not be only approach it should be well considered that during this entire process their was no heavy metal uptake by plants. For this it is better to address suggestion as mentioned in point 1 and if there was certain amount of heavy metal present it is important to also monitor the heavy metal uptake in plant. As Acacia is fast going plant it can bioaccumulate heavy metal in biomass.

Author: We put this information in our paper: “A treated sewage sludge from the “Estação de Tratamento de Esgoto (ETE)”, Juramento municipality, Minas Gerais State, Brazil, had no helminth eggs and protozoan cysts, and did not increase the heavy metal contents in grains of Zea mays L. (Poales: Poaceae) and Vigna unguiculata (L.) Walp. (Fabales: Fabaceae) [49].”

5. It may be the case that for this particular study sewage sludge is devoid of any heavy metal concentration but those who would like to replicate this approach it is important that this caution is well discussed in the discussion section.

Author: We put this information in our paper: “A treated sewage sludge from the “Estação de Tratamento de Esgoto (ETE)”, Juramento municipality, Minas Gerais State, Brazil, had no helminth eggs and protozoan cysts, and did not increase the heavy metal contents in grains of Zea mays L. (Poales: Poaceae) and Vigna unguiculata (L.) Walp. (Fabales: Fabaceae) [49].”

---

## [Editor Report · Decision Letter 1]

23 Jul 2020

Ecological indices of phytophagous Hemiptera and their natural enemies on Acacia auriculiformis (Fabales: Fabaceae) plants with or without dehydrated sewage sludge application in a degraded area

PONE-D-20-08541R1

Dear Dr.Demolin Leite

We’re pleased to inform you that your manuscript has been judged scientifically suitable for publication and will be formally accepted for publication once it meets all outstanding technical requirements.

Kind regards,

Tunira Bhadauria, Ph.D.

Academic Editor

PLOS ONE

---

## [Editor Report · Acceptance letter]

29 Jul 2020

PONE-D-20-08541R1 

Ecological indices of phytophagous Hemiptera and their natural enemies on Acacia auriculiformis (Fabales: Fabaceae) plants with or without dehydrated sewage sludge application in a degraded area 

Dear Dr. Leite:

I'm pleased to inform you that your manuscript has been deemed suitable for publication in PLOS ONE. Congratulations! Your manuscript is now with our production department. 

Kind regards, 

on behalf of

Dr. Tunira Bhadauria 

Academic Editor

PLOS ONE